# Study of the Association between *VEGF* Polymorphisms and the Risk of Coronary Artery Disease in Koreans

**DOI:** 10.3390/jpm12050761

**Published:** 2022-05-07

**Authors:** Eun-Ju Ko, In-Jai Kim, Jeong-Yong Lee, Hyeon-Woo Park, Han-Sung Park, Sang-Hoon Kim, Jae-Youn Moon, Jung-Hoon Sung, Nam-Keun Kim

**Affiliations:** 1Department of Biomedical Science, College of Life Science, CHA University, Seongnam 13488, Korea; ejko05@naver.com (E.-J.K.); smilee3625@naver.com (J.-Y.L.); aabb1114@naver.com (H.-W.P.); hahnsung@naver.com (H.-S.P.); 2Department of Cardiology, CHA Bundang Medical Center, CHA University, Seongnam 13496, Korea; mdij24@chol.com (I.-J.K.); kimsnag97@cha.ac.kr (S.-H.K.); answodus@cha.ac.kr (J.-Y.M.)

**Keywords:** coronary artery disease, *VEGF*, promoter region, 3′-untranslated region, endothelial homeostasis, angiogenesis

## Abstract

Coronary artery disease (CAD), a leading cause of death worldwide, has a complex etiology comprising both traditional risk factors (type 2 diabetes, dyslipidemia, arterial hypertension, and cigarette smoking) and genetic factors. Vascular endothelial growth factor (VEGF) notably contributes to angiogenesis and endothelial homeostasis. However, little is known about the relationship between CAD and *VEGF* polymorphisms in Koreans. The aim of this study is to investigate the associations of 2 *VEGF* promoter region polymorphisms (−1154G>A [rs1570360], −1498T>C [rs833061]) and 4 *VEGF* 3′-UTR polymorphisms (+936C>T [rs3025039], +1451C>T [rs3025040], +1612G>A [rs10434], and +1725G>A [rs3025053]) with CAD susceptibility in Koreans. We studied 885 subjects: 463 CAD patients and 422 controls. Genotyping was conducted with polymerase chain reaction-restriction fragment length polymorphism analysis and TaqMan allelic discrimination assays, and the genotype frequencies were calculated. We then performed haplotype and genotype combination analyses and measured the associations between *VEGF* polymorphisms and clinical variables in both the CAD patients and control subjects. We detected statistically significant associations between CAD and certain *VEGF* allele combinations. In the haplotypes of 5 single-nucleotide polymorphisms, the *VEGF* allele combination −1154A/+936T was associated with a decreased prevalence of CAD (A-T-T-G-G of *VEGF* −1154G>A/−1498T>C/+936C>T/+1612G>A/+1725G>A, AOR = 0.077, *p* = 0.021). In contrast, the *VEGF* allele combinations −1498T/+1725A and −1498T/+1612A/+1725A were associated with an increased prevalence of CAD (G-T-C-C-A of *VEGF* −1154G>A/−1498T>C/+936C>T/+1451C>T/+1725G>A, AOR = 1.602, *p* = 0.047; T-C-C-A-A of *VEGF* −1498T>C/+936C>T/+1451C>T/+1612G>A/+1725G>A, AOR = 1.582, *p* = 0.045). Gene–environment combinatorial analysis showed that the combination of the *VEGF* +1725AA genotype and several clinical factors (e.g., body mass index, hemoglobin A1c, and low-density lipoprotein cholesterol) increased the risk of CAD. Therefore, we suggest that *VEGF* polymorphisms and clinical factors may impact CAD prevalence.

## 1. Introduction

Coronary artery disease (CAD) is a major cause of global morbidity and mortality despite advanced prevention and treatment strategies. In CAD, the coronary arteries—the vessels that supply blood and oxygen to the heart—are narrowed or blocked by a thrombus or spasm [1]. CAD is divided into non-obstructive coronary atherosclerosis, stable angina pectoris, unstable angina pectoris, and acute myocardial infarction based on clinical symptoms, including the extent of arterial blockage and myocardial damage [2].

Atherosclerosis, characterized by plaque build-up in the vascular endothelium, is a major cause of CAD. Platelet activation and thrombus formation occur when the plaque is disrupted (e.g., by rupture, fissure, or ulceration), eventually leading to the narrowing of blood vessels or thrombotic obstruction [3]. Atherosclerosis was previously considered a cholesterol accumulation disease but is now understood to be caused by complex interactions among its risk factors, which include hypertension (HTN) [4], tobacco use [5], dyslipidemia [6], diabetes mellitus (DM) [7], and obesity [8] that change molecular messages in the blood and cells of the artery wall. Chronic exposure to these risk factors and other triggers weaken the vascular endothelium’s defense mechanisms, undermining its integrity and causing endothelial cell dysfunction [9], which is considered the earliest marker of atherosclerosis and a key variable in its pathogenesis and complications [10,11,12].

Genetic risk is estimated to account for 40~60% of susceptibility to CAD [13]. Since 2007, genome-wide association studies (GWAS) have found significant associations between 321 chromosomal loci and CAD. The identified chromosomal loci were related to blood pressure, lipid metabolism, adiposity, insulin resistance, neovascularization and angiogenesis, immune response and inflammation, NO-signaling, thrombosis, vascular remodeling [14]. Neovascularization and vascular remodeling are associated with lesion formation, and *VEGF* is one of the genes associated with neovascularization and angiogenesis [15].

The *VEGF* gene—located on chromosome 6p12–p21.1 [16] and comprising 8 exons and 7 introns [17]—gives rise to multiple vascular endothelial growth factor (VEGF) isoforms (e.g., VEGF121, VEGF145, VEGF165, VEGF189, VEGF206) via alternative exon splicing and various post-transcriptional mechanisms [18]. VEGF is an essential component of angiogenesis due to its pro-angiogenic activity and has several additional functions, including stimulating vascular permeability [19] and endothelial cell proliferation, promoting cell migration [20], and inhibiting apoptosis [21]. Furthermore, VEGF is one of the most important factors in maintaining endothelial cell homeostasis [22].

VEGF has been implicated in the pathogenesis of several diseases [23,24]. For example, intravitreal VEGF levels are associated with the development of diabetic retinopathy [25,26]; VEGF enhances leukocyte adhesion and stimulates the generation of reactive oxygen species to cause endothelial dysfunction, making it a key regulator of ocular angiogenesis. VEGF is also overexpressed in tumor cells, where it stimulates cell proliferation and survival and induces blood vessel formation to promote tumor growth [27,28]. Expression of VEGF is up-regulated by hypoxia, inflammation, wound-healing, and other pathological processes [29]. Especially, one study showed that circulating levels of total VEGF-A and VEGF-A165b in CAD patients were associated with syntax score, indicating the severity and complexity of CAD [30], and another study showed that VEGF 165b induced a neovascular response in the adventitia, and enhanced intimal thickening through the peri-adventitial collar placement [31]. Cai et al. [32] demonstrated that endothelial progenitor cell proliferation mediated by VEGF and IL-8 secretion is related to cardiac shock wave therapy, and Song et al. [33] showed that VEGF, derived from transplanted bone marrow-mesenchymal stem cell, regulated the expression of miRNAs such as miRNA-23a and miRNA-92a and performed anti-apoptotic effects in cardiomyocytes after MI.

*VEGF* has functional single-nucleotide polymorphisms (SNPs) in its promoter region and, 5′- and 3′- untranslated regions (UTRs) that may affect VEGF expression [34,35]. Some research has reported that several *VEGF* SNPs alter the level of VEGF expression and are associated with various diseases [36,37].

The −1154G>A polymorphism (rs1570360) in the promoter region of *VEGF* is associated with HTN-related chronic kidney disease [38], rheumatoid arthritis [39], and carotid artery stenosis [40]. The −1498T>C (rs833061) polymorphism, also located in the promoter region, is associated with susceptibility to breast cancer [41] and colorectal cancer [42]. Moreover, the −1498T>C affected the therapeutic effects of anti-angiogenic drugs targeting VEGF signaling cascades in renal cell carcinoma and hypertension [43]. These promoter region polymorphisms (e.g., −1154 and −1498) modulate the VEGF transcription level by altering promoter activity and are associated with hepatocellular carcinoma [44,45]. Of the 3′-UTR polymorphisms (+936C>T, +1451C>T, +1612G>A, and +1725G>A), the +936C>T polymorphism is associated with diabetic retinopathy, polycystic ovarian syndrome, and certain types of cancer, such as glioma, leukemia, and breast cancer [46,47,48,49,50]. In Koreans specifically, the +1451C>T polymorphism is associated with colorectal cancer susceptibility [51] and the +1612G>A and +1725G>A polymorphisms are associated with recurrent pregnancy loss [52].

The −1154G>A and −1498T>C polymorphisms in the promoter region have been studied in CAD patients of other populations [53], but not in CAD patients in Korea. Moreover, among the 3′-UTR region polymorphisms, +1451C>T, +1612G>A, and +1725G>A, excluding +936C>T, have not been studied in CAD. Therefore, we selected two *VEGF* promoter region polymorphisms (−1154G>A [rs1570360], −1498T>C [rs833061]) and four *VEGF* 3′-UTR polymorphisms (+936C>T [rs3025039], +1451C>T [rs3025040], +1612G>A [rs10434], +1725G>A [rs3025053]) to investigate the association between *VEGF* and CAD in the Korean population.

## 2. Materials and Methods

### 2.1. Study Population

All study protocols for this genetic analysis were reviewed and approved by The Institutional Review Board of CHA Bundang Medical Center of CHA University (IRB number: 2013-10-114) and followed the recommendations of the Declaration of Helsinki (Fifth revision, 7 October 2000). Written informed consent was obtained from all study participants prior to enrollment.

The study subjects were recruited from the South Korean provinces of Seoul and Gyeonggi-do between 2014 and 2016. Patients were referred from the Department of Cardiology at the CHA Bundang Medical Center during this period; those who presented with stable CAD or acute coronary syndromes—including unstable angina with or without ST-segment elevation—and at least one coronary lesion with >50% stenosis in a vessel with a diameter of 2.25–4.00 mm were screened for eligibility. No restrictions were placed on the number or length of treated lesions, which vessels were treated, or the number of implanted stents. The exclusion criteria were acute myocardial infarction and a life expectancy less than one year. The final sample size was 463 CAD patients, all of whom underwent electrocardiography as well as coronary angiography for diagnosis, the results of which were confirmed by at least one independent experienced cardiologist.

We randomly selected 422 sex- and age-matched control subjects from patients who presented at the Department of Cardiology at the CHA Bundang Medical Center during the same period for health examinations, including biochemical testing, electrocardiography, coronary computed tomography, and brain magnetic resonance imaging. The control subjects did not have a history of myocardial infarction or a recent history of angina symptoms, and the exclusion criteria were the same as those used for the patient group.

We established the following definitions of CAD risk factors for this study: HTN—systolic blood pressure (BP) >130 mmHg and diastolic BP >85 mmHg (included patients currently taking hypertensive medications); DM—fasting plasma glucose level >110 mg/dL (included patients taking diabetic medications); smoking—current cigarette smoking; hyperlipidemia—fasting serum total cholesterol level ≥150 mg/dL or a history of anti-hyperlipidemia agent treatment.

### 2.2. Estimation of Biochemical Factor Concentrations

Peripheral blood samples were collected in blood collection tubes containing an anticoagulant 12 h after each patient’s previous meal. The samples were centrifuged for 15 min at 1000× *g* to separate plasma from whole blood. Plasma homocysteine concentrations were measured using an IMx fluorescence polarizing immunoassay (Abbott Laboratories, Abbott Park, IL, USA), and plasma folate concentrations were measured with a radioassay kit (ACS:180; Bayer, Tarrytown, NY, USA). The levels of total cholesterol, triglycerides (TG), high-density lipoprotein cholesterol (HDL-cholesterol), and low-density lipoprotein cholesterol (LDL-cholesterol) were determined by enzymatic colorimetric methods using commercial reagent sets (TBA 200FR NEO, Toshiba Medical Systems, Tokyo, Japan).

### 2.3. Genotyping

Peripheral blood samples were collected in blood collection tubes and were treated with ethylene-diamine-tetraacetic acid (EDTA). The samples were centrifuged at 3000 rpm for 15 min, the buffy coat layer was collected, and the leukocytes were separated. DNA was extracted from the subjects’ leukocytes using a G-DEX II Genomic DNA Extraction Kit (Intron Biotechnology, Seongnam, Korea), according to the manufacturer’s instructions. We used polymerase chain reaction–restriction fragment length polymorphism (PCR-RFLP) to analyze *VEGF* −1154G>A and +936C>T. PCR was performed using the 2× h-Taq PCR PreMix (Solgent Corporation, Daejeon, Korea). *VEGF* −1498G>A, +1451C>T, +1612G>A, +1725G>A were genotyped by real-time PCR (RG-6000, Corbett Research, Australia) for allelic discrimination. *VEGF* +1451C>T, +1612C>T, and +1725G>A primers as well as TaqMan probes were designed using Primer Express Software (v2.0) and synthesized by Bioneer (Daedeok-gu, Daegu, Korea) with the FAM and JOE reporter dyes. Real-time PCR was performed using the 2× Real-time Smart mix (Solgent Corporation, Daejeon, Korea). *VEGF* −1498G>A was genotyped using a TaqMan SNP Genotyping Assay Kit (Applied Biosystems, Foster City, CA, USA). Genotyping was performed according to the manufacturer’s protocols. The primers, probe sequences, and PCR conditions are described in Appendix A. The PCR conditions for *VEGF* −1154G>A and +936C>T followed these steps: (i) pre-denaturation at 95 °C for 15 min, (ii) 35 cycles of denaturation at 95 °C for 30 s, (iii) annealing at each optimized temperature for 30 s, (iv) extension at 72 °C for 30 s, (v) and final extension at 72 °C for 5 min. Restriction enzyme digestions were performed at 37 °C for 16 h using *Mnl*Ⅰ for −1154G>A and *Nla*Ⅲ for +936C>T, and the enzymes were inactivated by incubation at 65 °C for 20 min. We randomly selected 10–15% of the PCR assays for each *VEGF* polymorphism and confirmed the results with automated DNA sequencing (ABI 3730xl DNA analyzer, Applied Biosystems, Foster City, CA, USA). The concordance of the quality control samples was 100%.

### 2.4. Statistical Analysis

To compare the clinical characteristics between the study groups, we used the Chi-square test for categorical data. Moreover, we checked the statistical normality of the continuous variables by confirming that they matched the normal distribution through the Kolmogorov–Smirnov test. The continuous variables showed non-normal distributions (*p* < 0.05 in Kolmogorov-Smirnov test), and the Mann-Whitney test was performed for the group difference analyses. The associations between the *VEGF* polymorphisms and CAD occurrence were calculated using adjusted odds ratios (AORs) and 95% confidence intervals (CIs) from a multivariate logistic regression adjusted for age, gender, HTN, DM, hyperlipidemia, and smoking status. The genotype distribution for each polymorphism was assessed for Hardy–Weinberg equilibrium (HWE) deviations. *p* values < 0.05 were considered statistically significant. Calculation of the FDR-*p* is a way to address the problems associated with multiple comparisons and provides a measure of the expected proportion of false-positives among data. These analyses were performed using Prism 4.0 (GraphPad Software Inc., San Diego, CA, USA) and Medcalc version 18.2.1 (Medcalc Software, Mariakerke, Belgium). HAPSTAT 3.0 (University of North Carolina, Chapel Hill, NC, USA) was used to estimate the haplotype frequencies of the *VEGF* polymorphisms. The Chi-square test and Fisher’s exact test were used for haplotype analysis.

## 3. Results

### 3.1. Clinical Profiles of Study Subjects

Table 1 presents the clinical variables of the 463 CAD patients and 422 control subjects. The case–control comparisons revealed no statistically significant differences between ages or genders (*p* = 0.498 and *p* = 0.425, respectively). The mean BMI of the CAD patients (mean ± standard deviation, 24.2 ± 3.3) was significantly higher than that of the control subjects (25.1 ± 3.4). HTN, DM, and metabolic syndrome (MetS), major risk factors for CAD, were more frequent in the patient group than in the control group (*p* < 0.05). No statistically significant difference in hyperlipidemia or smoking status was found between the two groups (*p* > 0.05). Total cholesterol, triglyceride, HDL-cholesterol, folate, creatinine were significantly different between the CAD and control groups (*p* < 0.05). LDL-cholesterol (*p* = 0.312), homocysteine (*p* = 0.312), vitamin B12 (*p* = 0.097) were not significantly different between the CAD and control groups.

### 3.2. Comparison of Genotype Frequencies of VEGF Polymorphisms

Table 2 presents the genotype frequencies of the six polymorphisms (*VEGF* −1154G>A [rs1570360], −1498T>C [rs833061], +936C>T [rs3025039], +1451C>T [rs3025040], +1612G>A [rs10434], +1725G>A [rs3025053]) in the CAD patients and control subjects. The allele frequencies of the associated SNP in other populations, according to 1000 genomes data, were compared with the control and CAD patient groups (Appendix A). Each genotype distribution was confirmed to be in HWE (*p* > 0.05). There was no statistically significant difference in the genotype frequencies of the *VEGF* polymorphic loci between the control and CAD groups (Table 2). Genotype analysis of the MetS subgroup investigated whether the associations of the six polymorphisms changed according to the presence or absence of MetS (Table 3). The CT and dominant models of the *VEGF* +936C>T polymorphism were associated with decreased CAD susceptibility in CAD patients with MetS (CC vs. CT, AOR = 0.643, 95% CI = 0.417–0.991, *p* = 0.045; CC vs. CT + TT AOR = 0.633, 95% CI = 0.415–0.965, *p* = 0.034).

Genotype analysis of the gender subgroup, which evaluated whether the genotype frequency changes according to gender, revealed that the *VEGF* +936TT, +1451TT genotypes were less frequent in female CAD patients than in the female control group (*VEGF* +936CC vs. CT + TT, AOR = 0.593, 95% CI = 0.399–0.881, *p* = 0.010; *VEGF* +1451CC vs. CT + TT, AOR = 0.617, 095% CI = 0.413–0.922, *p* = 0.018; see Appendix A).

### 3.3. Haplotype Analysis and Genotype Combination Analysis

Haplotype analysis and genotype combination analysis were performed to confirm the combined effect of the six SNPs (Table 4 and Table 5). The haplotype analysis of *VEGF* −1154G>A/−1498T>C/+936C>T/+1451C>T/+1612G>A/+1725G>A showed that G-T-T-C-G-G (AOR = 0.056, *p* = 0.004), G-C-C-C-A-A (AOR = 0.056, *p* = 0.004), and A-T-C-C-G-G (AOR = 0.035, *p* < 0.001) were associated with CAD susceptibility. Among the haplotypes of five SNPs, −1154A allele, +936T allele were associated with a decreased prevalence of CAD (A-T-C-C-G of *VEGF* −1154G>A/−1498T>C/+936C>T/+1451C>T/+1725G>A, AOR = 0.036, *p* < 0.0001; T-T-C-G-G of *VEGF* −1498T>C/+936C>T/+1451C>T/+1612G/A+1725G>A, AOR = 0.039, *p* = 0.001). The *VEGF* allele combination −1154A/+936T was also associated with a decreased prevalence of CAD (A-T-T-G-G of *VEGF* −1154G>A/−1498T>C/+936C>T/+1612G>A/+1725G>A, AOR = 0.077, *p* = 0.021). In contrast, the *VEGF* allele combinations −1498T/+1725A and −1498T/+1612A/+1725A were associated with an increased prevalence of CAD (G-T-C-C-A of *VEGF* −1154G>A/−1498T>C/+936C>T/+1451C>T/+1725G>A, AOR = 1.602, *p* = 0.047; T-C-C-A-A of *VEGF* −1498T>C/+936C>T/+1451C>T/+1612G>A/+1725G>A). Similarly, among the haplotypes of four SNPs, the *VEGF* allele combination −1154A/+936T was more frequent in the control group (A-T-T-G of *VEGF* −1154G>A/−1498T>C/+936C>T/+1612G>A, AOR = 0.076, *p* = 0.020; A-T-T-G of *VEGF* −1154G>A/−1498T>C/+936C>T/+1725G>A, AOR = 0.067, *p* = 0.010). The *VEGF* allele combinations −1498T/+1612A and −1498T/+1612A/+1725A were more frequent in the CAD group (G-T-C-A of *VEGF* −1154G>A/−1498T>C/+936C>T/+1725G>A, AOR = 1.649, *p* = 0.049; T-C-A-A of *VEGF* −1498T>C/+936C>T/+1612G>A/+1725G>A, AOR = 1.678, *p* = 0.025). These trends were also observed in three *VEGF* polymorphisms during the haplotype analysis (A-T-C of *VEGF* −1154G>A/−1498T>C/+936C>T, AOR = 0.070, *p* < 0.001; T-C-A of *VEGF* −1498T>C/+936C>T/+1725G>A, AOR = 1.829, *p* = 0.010; T-A-A of *VEGF* −1498T>C/+1612G>A/+1725G>A, AOR = 1.660, *p* = 0.024; Appendix A).

Genotype combination analysis was performed to confirm combined genotype effect of the six SNPs. Prior to genotype combination analysis, MDR was performed on the six SNPs to identify interactions that influence CAD risk. The three SNPs (*VEGF* −1154G>A, −1498T>C, and +936C>T) were selected as the best MDR model. In *VEGF* −1154G>A/−1498T>C/+936C>T genotype combination analysis, the combined genotype of VEGF −1154GA/−1498TT/+936CC and *VEGF*−1154GA/−1498TC/+936TT results that indicated an association with CAD risk (AOR = 0.224, *p* = 0.027; AOR = 0.230, *p* = 0.048, respectively, Appendix A). Appendix A shows the results of the combined genotype analysis. Four genotypes were less frequent in patients with CAD than in controls, and these reduced frequencies were associated with lower susceptibility to CAD. Specifically, these include the *VEGF* −1154G>A/−1498T>C combination GG/CC and GA/TT, the *VEGF*−1154G>A/+1612G>A combination GA/GG, the *VEGF* +936C>T/+1451C>T combination CT/CC. In contrast, *VEGF* +1612G>A/+1725G>A combination AA/GA genotype was associated with increased risk of CAD.

The linkage disequilibrium analysis further confirmed that *VEGF* −1154G>A/−1498C>T and *VEGF* +936C>T/+1451C>T/+1612G>A/+1725G>A were in strong disequilibrium between the CAD patients and control subjects (Figure 1).

### 3.4. Synergistic Effect of VEGF Polymorphisms and Clinical Factors

We then analyzed the synergistic effect of the *VEGF* polymorphisms and clinical factors (i.e., HTN, DM, hemoglobin A1c, smoking, BMI, hyperlipidemia, MetS, folate, and homocysteine) on CAD risk (Appendix A). Our results show that *VEGF* +1612G>A and LDL-cholesterol have a synergistic effect for increased susceptibility of CAD (GA+AA, AOR = 8.278, 95% CI = 3.640–18.827, *p* < 0.0001). Moreover, combining the *VEGF* +1725 GA + AA type with HbA1c (AOR = 7.099, 95% CI = 2.269–22.212, *p* = 0.001), BMI (AOR = 3.340, 95% CI = 1.650–6.761, *p* = 0.001), and LDL-cholesterol (AOR = 10.962, 95% CI = 3.633–33.078, *p* < 0.0001) resulted in an increased CAD risk.

### 3.5. Clinical Variables in CAD Patients by VEGF Polymorphism Status

We conducted a one-way analysis of variance (ANOVA) between the genes and clinical factors of all participants and subgroups (Appendix A). The HDL-cholesterol and vitamin B12 levels of all participants were significantly associated with *VEGF* +1451C>T (*p* = 0.029 and *p* = 0.050, respectively). The homocysteine levels of all subjects were significantly associated with *VEGF* +1725G>A (*p* = 0.017), and those of the CAD patients were significantly associated with *VEGF* +1725G>A (*p* = 0.033, Appendix A). The creatinine levels of the control group showed a decreasing trend depending on *VEGF* +936C>T (*p* = 0.004, Appendix A). In contrast, systolic BP demonstrated an increasing trend dependent on *VEGF* +1451C>T in the control group (*p* = 0.014, Appendix A).

## 4. Discussion

The coronary arteries are blood vessels that branch off from the aorta and supply oxygen and nutrients to the heart muscle. CAD—a major cause of mortality worldwide as well as in Korea—results from the narrowing or blockage of the coronary arteries by blood clots or vasospasms, reducing blood supply to the heart muscle [19,54]. CAD is a multifactorial disease affected by several risk factors, both environmental (e.g., DM, HTN, hyperlipidemia, smoking, alcohol intake, obesity) [55] and genetic [56,57,58]. However, not all people exposed to these environmental factors develop CAD, which points to the role of genetic factors.

Vascular endothelial cells, located on the intima (the inner lining of the vasculature), serve as the main regulators of vasodilation, vasoconstriction, thrombosis, thrombolysis, and the growth and migration of vascular smooth muscle cells [59]. Endothelial dysfunction predicts cardiovascular outcomes and plays an important role in the development of atherosclerosis [60,61] by disrupting vascular homeostasis and causing damage to the arterial walls that can lead to fibrosis proliferation with inflammatory reactions. These events can serve as an early marker of atherosclerosis [11].

VEGF is associated with endothelial cell dysfunction. Previous studies have reported that vascular growth factors (such as VEGF) affect the survival of endothelial cells through regulation of metabolism and autophagy [62]. Suppressing VEGF results in vascular toxicity [63] and increased apoptosis [64]. In particular, VEGF expression levels were decreased in CAD patients [65] and showed the linear association with MetS score (0–5) in CAD patients [66]. Other studies showed that miR-214/361-5p were able to suppress VEGF expression and endothelial progenitor cell angiogenic activity [67,68]. Moreover, *VEGF* gene therapy is one of the safe and effective treatments in case of a cardiac event [69]. Furthermore, atorvastatin is one of the treatments for dyslipidemia and has been demonstrated to prevent cardiovascular disease. Sun et al. [70] indicated atorvastatin down-regulated miR-221 and increased VEGA protein levels, and improved the cell proliferation, migration, and EPC angiogenesis activity.

In the present study, we investigated the association between CAD susceptibility and six *VEGF* polymorphisms—two in the promoter region (−1154G>A [rs1570360] and −1498T>C [rs833061]) and four in the 3′-UTR (+936C>T [rs3025039], +1451C>T [rs3025040], +1612G>A [rs10434], and +1725G>A [rs3025053]). These polymorphisms may act as one functional polymorphism to regulate both VEGF expression levels and mRNA stability.

The results of our genotype analysis showed no significant differences in CAD prevalence for the *VEGF* polymorphisms. Case-control studies have suggested that the −1154G>A, +936C>T polymorphisms of *VEGF* are associated with susceptibility to CAD [71]. However, the −1154G>A, +936C>T polymorphisms have not been associated with CAD in other studies [72,73]. Clearly, the results of previous studies between −1154G>A, +936C>T polymorphism and CAD are controversial. However, in patients with MetS, *VEGF* +936TT and the CT+TT genotype were found to be associated with decreased CAD risk. In female CAD patients, *VEGF* +936TT, the CT+TT genotype, and *VEGF* +1451CT+TT were associated with decreased CAD risk. Gender difference in the genetic risk of CAD may be more confounded by environmental/lifestyle risk factors [74], and similarly, another study showed that in women the *VEGF* -2578C/+634C/+936C haplotype was found to be associated with higher carotid IMT, but in men no similar association was found or it was even in the opposite direction [75]. Recently, one study performed sex-stratified GWAS, identifying ten variations that were associated with CAD, and nine of the ten had stronger effects in male participants. Conversely, MYOZ2 rs7696877 polymorphism showed significant effect in females [14]. Our haplotype analysis revealed that the *VEGF* allele combination −1154A/+936T was associated with a decreased risk of CAD. In contrast, the *VEGF* allele combination −1498T/+1612A/+1725A was associated with an increased risk of CAD. *VEGF* +1612G>A and +1725G>A, combined with the clinical factors LDL-cholesterol, HbA1c, and BMI, had a synergistic effect on CAD prevalence resulting in increased risk. Furthermore, homocysteine levels were associated with the *VEGF* +1725G>A polymorphism. These results demonstrated for the first time that haplotypes of *VEGF* six polymorphisms and genotype combinations are associated with the occurrence of CAD, and that interactions between *VEGF* polymorphisms and other clinical factors are associated with the prevalence of CAD. However, further studies are needed to investigate the underlying mechanisms.

Polymorphisms in the *VEGF* promoter region may alter transcription factor binding and affect mRNA translation and the expression of proteins. In one case, the *VEGF* −1498CT/CC genotype was associated with higher mRNA levels of VEGF in peripheral blood, and the luciferase reporter assay demonstrated that the *VEGF* −1498C allele increased transcription activity to a greater extent than the T allele [76]. Furthermore, another study reported that the *VEGF* −2578/−1154/−1498 haplotype may change the *VEGF* gene expression in myoblast cells in a hypoxic environment, and *VEGF* promoter polymorphisms may affect the VEGF angiogenesis function in pathological conditions [77].

The 3′-UTR plays several roles in the regulation of mRNA processes, such as localization, stability, and translation [78]. The 3′ UTR is a miRNA binding site [79]; thus, 3′-UTR polymorphisms may have a considerable impact on gene expression by abolishing, weakening, or creating miRNA binding sites. For example, one study reported that the *VEGF* +936C>T, +1451C>T haplotype and genotype produced higher plasma VEGF levels and mRNA stability in chronic kidney disease [45]. In another study that used a luciferase reporter assay, when a T allele was present in the *VEGF* 3′-UTR +1451C>T polymorphism, it interacted with miRNA-199a and -199b and reduced the expression of luciferase [80].

This study has several limitations. First, the manner in which the *VEGF* promoter region and 3′-UTR polymorphisms affect the development of CAD remains unclear. The effects of *VEGF* SNPs should also be confirmed using in vitro studies. Second, there is a lack of information regarding additional environmental risk factors for CAD. Lastly, the sample sizes of both the CAD patients and control subjects were small and the study population included only Korean individuals. Although the results of our study first reported associations between *VEGF* 3′-UTR polymorphisms (+1612G>A, +1725G>A) and CAD susceptibility, prospective studies should validate our results using larger sample sizes and other ethnic groups.

## 5. Conclusions

We investigated six *VEGF* polymorphism frequencies (−1154G>A, −1498T>C, +936C>T, +1451C>T, +1612G>A, and +1725G>A) between CAD patients and control subjects. We found that the *VEGF* haplotypes −1154G>A/+936C>T and −1498C>T/+1612G>A/+1725G>A and the genotype combinations −1154G>A/−1498C>T, +936C>T/+1451C>T, and +1612G>A/+1725G>A correlated with the occurrence and interaction of *VEGF* polymorphisms +1612G>A and +1725G>A and that some clinical factors may increase the risk of CAD.

## Figures and Tables

**Figure 1 jpm-12-00761-f001:**
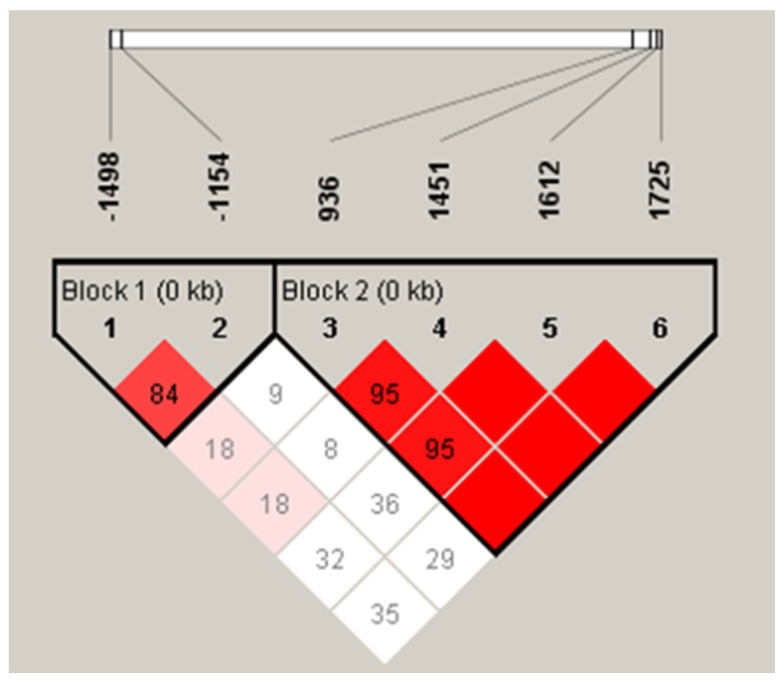
Linkage disequilibrium between *VEGF* loci. There was strong LDs between loci −1498T>C and −1154 (D’ = 0.848), +936C>T and +1451C>T (D’ = 0.957), +936C>T and +1612G>A (D’ = 0.956), +936C>T and +1725G>A (D’ = 1.000), +1451C>T and +1612G>A (D’ = 1.000), +1451C>T and +1725G>A (D’ = 1.000), and +1612G>A and +1725G>A (D’ = 1.000).

**Table 1 jpm-12-00761-t001:** Baseline characteristics between CAD patients and control subjects.

Characteristic	Control Subjects (*n* = 422)	CAD Patients (*n* = 463)	*p*
Age (years, mean ± SD)	61.1 ± 11.8	61.6 ± 11.4	0.502
Male (%)	172 (40.8)	201 (43.4)	0.425
BMI (kg/m^2^, mean ± SD)	24.2 ± 3.3	25.1 ± 3.4	0.0001
Hypertension (*n*, %)	161 (38.2)	256 (56.1)	<0.0001
Diabetes mellitus (*n*, %)	53 (12.6)	126 (27.6)	<0.0001
Fasting blood sugar (mg/dL, mean ± SD)	114.1 ± 37.3	141.5 ± 63.0	<0.0001
Hyperlipidemia (*n*, %)	96 (22.7)	126 (27.4)	0.113
Total cholesterol (mg/dL, mean ± SD)	192.1 ± 37.4	186.8 ± 46.8	0.006
Triglyceride (mg/dL, mean ± SD)	146.2 ± 91.1	158.6 ± 108.4	0.018
HDL-cholesterol (mg/dL, mean ± SD)	46.7 ± 13.8	43.9 ± 11.2	0.016
LDL-cholesterol (mg/dL, mean ± SD)	117.3 ± 42.1	113.0 ± 40.0	0.312
Smoking (*n*, %)	137 (32.5)	141 (30.7)	0.608
Metabolic syndrome (*n*, %)	152 (36.0)	293 (63.3)	<0.0001
Homocysteine (μmol/L, mean ± SD)	9.8 ± 4.2	9.9 ± 5.3	0.312
Vitamin B12 (pg/mL, mean ± SD)	677.3 ± 264.3	663.4 ± 338.2	0.097
Folate (nmol/L, mean ± SD)	8.6 ± 7.3	8.7 ± 9.6	0.033
Creatinine (mg/dL, mean ± SD)	0.9 ± 0.2	1.5 ± 6.7	0.0004

Note: CAD, coronary artery disease; BMI, body mass index; HDL, high-density lipoprotein; LDL, low-density lipoprotein. *p* was calculated using the Mann–Whitney test for continuous variables and Chi-square test for categorical variables.

**Table 2 jpm-12-00761-t002:** Comparison of genotype frequencies of *VEGF* polymorphisms between CAD patients and control subjects.

Genotype	Control Subjects (*n* = 422)	CAD Patients (*n* = 463)	COR (95% CI)	*p*	FDR-*p*	AOR (95% CI)	*p*	FDR-*p*
*VEGF* −1154G>A								
GG	293 (69.4)	340 (73.4)	1.000 (reference)			1.000 (reference)		
GA	121 (28.7)	112 (24.2)	0.798 (0.591–1.078)	0.141	0.846	0.823 (0.603–1.123)	0.219	0.778
AA	8 (1.9)	11 (2.4)	1.185 (0.470–2.985)	0.719	0.719	1.073 (0.405–2.846)	0.887	0.887
Dominant (GG vs. GA + AA)			0.822 (0.613–1.101)	0.188	0.867	0.838 (0.619–1.134)	0.252	0.648
Recessive (GG + GA vs. AA)			1.259 (0.502–3.162)	0.623	0.623	1.111 (0.420–2.939)	0.833	0.833
HWE-*p*	0.264	0.624						
*VEGF* −1498T>C								
TT	241 (57.1)	267 (57.7)	1.000 (reference)			1.000 (reference)		
TC	158 (37.4)	175 (37.8)	1.000 (0.758–1.319)	0.999	0.999	0.935 (0.701–1.248)	0.648	0.778
CC	23 (5.5)	21 (4.5)	0.824 (0.445–1.527)	0.539	0.647	0.783 (0.405–1.513)	0.467	0.584
Dominant (TT vs. TC + CC)			0.977 (0.749–1.276)	0.867	0.867	0.919 (0.696–1.214)	0.554	0.665
Recessive (TT + TC vs. CC)			0.824 (0.449–1.512)	0.532	0.623	0.810 (0.425–1.541)	0.520	0.651
HWE-*p*	0.660	0.252						
*VEGF* +936C>T								
CC	273 (64.7)	311 (67.2)	1.000 (reference)			1.000 (reference)		
CT	132 (31.3)	137 (29.6)	0.911 (0.682–1.217)	0.528	0.999	0.849 (0.629–1.147)	0.286	0.778
TT	17 (4.0)	15 (3.2)	0.775 (0.380–1.580)	0.483	0.647	0.716 (0.334–1.535)	0.391	0.584
Dominant (CC vs. CT + TT)			0.896 (0.678–1.183)	0.437	0.867	0.837 (0.626–1.110)	0.228	0.648
Recessive (CC + CT vs. TT)			0.798 (0.393–1.618)	0.531	0.623	0.781 (0.368–1.660)	0.521	0.651
HWE-*p*	0.835	0.985						
*VEGF* +1451C>T								
CC	283 (67.1)	315 (68.0)	1.000 (reference)			1.000 (reference)		
CT	119 (28.2)	133 (28.7)	1.004 (0.748–1.348)	0.978	0.999	0.930 (0.684–1.263)	0.640	0.778
TT	20 (4.7)	15 (3.2)	0.674 (0.339–1.341)	0.261	0.653	0.639 (0.306–1.334)	0.233	0.583
Dominant (CC vs. CT + TT)			0.957 (0.722–1.268)	0.758	0.867	0.892 (0.665–1.197)	0.448	0.665
Recessive (CC + CT vs. TT)			0.673 (0.340–1.332)	0.256	0.623	0.666 (0.322–1.374)	0.271	0.651
HWE-*p*	0.110	0.834						
*VEGF* +1612G>A								
GG	298 (70.6)	320 (69.1)	1.000 (reference)			1.000 (reference)		
GA	114 (27.0)	125 (27.0)	1.021 (0.757–1.377)	0.891	0.999	0.979 (0.717–1.336)	0.892	0.892
AA	10 (2.4)	18 (3.9)	1.676 (0.762–3.690)	0.199	0.653	1.880 (0.835–4.234)	0.127	0.583
Dominant (GG vs. GA + AA)			1.074 (0.806–1.432)	0.627	0.867	1.040 (0.771–1.403)	0.796	0.796
Recessive (GG + GA vs. AA)			1.667 (0.761–3.652)	0.202	0.623	1.778 (0.793–3.986)	0.162	0.651
HWE-*p*	0.816	0.195						
*VEGF* +1725G>A								
GG	377 (89.3)	403 (87.0)	1.000 (reference)			1.000 (reference)		
GA	45 (10.7)	57 (12.3)	1.185 (0.782–1.795)	0.423	0.999	1.176 (0.765–1.808)	0.460	0.778
AA	0 (0.0)	3 (0.6)	N/A	N/A	N/A	N/A	N/A	N/A
Dominant (GG vs. GA + AA)			1.247 (0.827–1.882)	0.292	0.867	1.239 (0.810–1.897)	0.324	0.648
Recessive (GG + GA vs. AA)			N/A	N/A	N/A	N/A	N/A	N/A
HWE-*p*	0.221	0.530						

CAD, coronary artery disease; COR, crude odds ratio; CI, confidence interval; AOR, adjusted odds ratio; FDR, false discovery rate. AOR: Adjusted by age, gender, hypertension, diabetes mellitus, hyperlipidemia, and smoking status.

**Table 3 jpm-12-00761-t003:** Comparison of genotype frequencies of *VEGF* polymorphisms between CAD patients and control subjects according to MetS.

Genotype	Control Subjects without MetS (*n* = 270)	CAD Patients without MetS (*n* = 170)	AOR (95% CI)	*p*	FDR-*p*	Control Subjects with MetS (*n* = 152)	CAD Patients with MetS (*n* = 293)	AOR (95% CI)	*p*	FDR-*p*
*VEGF* −1154G>A										
GG	189 (70.0)	119 (70.0)	1.000 (reference)			104 (68.4)	221 (75.4)	1.000 (reference)		
GA	75 (27.8)	48 (28.2)	1.072 (0.689–1.669)	0.758	0.816	46 (30.3)	64 (21.8)	0.669 (0.425–1.054)	0.083	0.249
AA	6 (2.2)	3 (1.8)	0.564 (0.109–2.912)	0.494	0.951	2 (1.3)	8 (2.7)	1.850 (0.375–9.134)	0.450	0.563
Dominant (GG vs. GA + AA)			1.034 (0.670–1.596)	0.88	0.880			0.718 (0.461–1.118)	0.143	0.286
Recessive (GG + GA vs. AA)			0.554 (0.107–2.859)	0.481	0.938			2.070 (0.421–10.181)	0.371	0.543
*VEGF* −1498T>C										
TT	154 (57.0)	93 (54.7)	1.000 (reference)			87 (57.2)	174 (59.4)	1.000 (reference)		
TC	100 (37.0)	69 (40.6)	1.106 (0.728–1.680)	0.638	0.816	58 (38.2)	106 (36.2)	0.857 (0.561–1.308)	0.475	0.531
CC	16 (5.9)	8 (4.7)	0.647 (0.238–1.759)	0.393	0.951	7 (4.6)	13 (4.4)	0.881 (0.326–2.382)	0.802	0.802
Dominant (TT vs. TC + CC)			1.041 (0.695–1.559)	0.846	0.880			0.862 (0.574–1.296)	0.476	0.476
Recessive (TT + TC vs. CC)			0.604 (0.224–1.632)	0.320	0.938			0.946 (0.359–2.492)	0.911	0.911
*VEGF* +936C>T										
CC	181 (67.0)	108 (63.5)	1.000 (reference)			92 (60.5)	203 (69.3)	1.000 (reference)		
CT	77 (28.5)	53 (31.2)	1.054 (0.677–1.640)	0.816	0.816	55 (36.2)	84 (28.7)	0.643 (0.417–0.991)	0.045	0.249
TT	12 (4.4)	9 (5.3)	1.053 (0.406–2.729)	0.916	0.951	5 (3.3)	6 (2.0)	0.529 (0.149–1.880)	0.325	0.542
Dominant (CC vs. CT + TT)			1.064 (0.698–1.620)	0.774	0.880			0.633 (0.415–0.965)	0.034	0.204
Recessive (CC + CT vs. TT)			1.188 (0.464–3.042)	0.720	0.938			0.609 (0.176–2.107)	0.434	0.543
*VEGF* +1451C>T										
CC	185 (68.5)	110 (64.7)	1.000 (reference)			98 (64.5)	205 (70.0)	1.000 (reference)		
CT	71 (26.3)	51 (30.0)	1.099 (0.701–1.724)	0.681	0.816	48 (31.6)	82 (28.0)	0.755 (0.485–1.176)	0.214	0.428
TT	14 (5.2)	9 (5.3)	0.972 (0.388–2.437)	0.951	0.951	6 (3.9)	6 (2.0)	0.441 (0.131–1.480)	0.185	0.463
Dominant (CC vs. CT + TT)			1.084 (0.709–1.658)	0.709	0.880			0.723 (0.471–1.111)	0.139	0.286
Recessive (CC + CT vs. TT)			1.011 (0.408–2.509)	0.981	0.981			0.507 (0.155–1.651)	0.259	0.543
*VEGF* +1612G>A										
GG	187 (69.3)	123 (72.4)	1.000 (reference)			111 (73.0)	197 (67.2)	1.000 (reference)		
GA	75 (27.8)	40 (23.5)	0.832 (0.525–1.318)	0.433	0.816	39 (25.7)	85 (29.0)	1.240 (0.780–1.971)	0.364	0.531
AA	8 (3.0)	7 (4.1)	1.221 (0.416–3.584)	0.717	0.951	2 (1.3)	11 (3.8)	4.022 (0.843–19.189)	0.081	0.405
Dominant (GG vs. GA + AA)			0.863 (0.557–1.337)	0.510	0.880			1.240 (0.780–1.971)	0.364	0.437
Recessive (GG + GA vs. AA)			1.192 (0.404–3.516)	0.750	0.938			3.614 (0.769–16.997)	0.104	0.520
*VEGF* +1725G>A										
GG	242 (89.6)	150 (88.2)	1.000 (reference)			135 (88.8)	253 (86.3)	1.000 (reference)		
GA	28 (10.4)	20 (11.8)	1.118 (0.595–2.102)	0.729	0.816	17 (11.2)	37 (12.6)	1.224 (0.651–2.300)	0.531	0.531
AA	0 (0.0)	0 (0.0)	N/A	N/A	N/A	0 (0.0)	3 (1.0)	N/A	N/A	N/A
Dominant (GG vs. GA + AA)			1.118 (0.595–2.102)	0.729	0.880			1.338 (0.717–2.499)	0.360	0.437
Recessive (GG + GA vs. AA)			N/A	N/A	N/A			N/A	N/A	N/A

CAD, coronary artery disease; MetS, Metabolic syndrome, COR, crude odds ratio; CI, confidence interval; AOR, adjusted odds ratio. AOR: Adjusted by age, gender, hypertension, diabetes mellitus, hyperlipidemia, and smoking status.

**Table 4 jpm-12-00761-t004:** Haplotype analysis of *VEGF* polymorphisms in CAD patients and controls subjects.

Haplotype	Control Subjects (2*n* = 844)	CAD Patients (2*n* = 926)	OR (95% CI)	*p*
*VEGF* −1154G>A/−1498T>C/+936C>T/+1451C>T/+1612G>A/+1725G>A				
G-T-C-C-G-G	399 (47.2)	475 (51.3)	1.000 (reference)	
G-T-T-C-G-G	7 (0.9)	0 (0.0)	0.056 (0.003–0.984)	0.004
G-C-C-C-A-A	7 (0.8)	0 (0.0)	0.056 (0.003–0.984)	0.004
A-T-C-C-G-G	24 (2.8)	1 (0.1)	0.035 (0.005–0.260)	<0.0001
*VEGF* −1154G>A/−1498T>C/+936C>T/+1451C>T/+1612G>A				
G-T-C-C-G	397 (47.0)	474 (51.2)	1.000 (reference)	
G-T-T-C-G	8 (0.9)	0 (0.0)	0.049 (0.003–0.857)	0.002
A-T-C-C-G	24 (2.9)	1 (0.2)	0.035 (0.005–0.259)	<0.0001
*VEGF* −1154G>A/−1498T>C/+936C>T/+1451C>T/+1725G>A				
G-T-C-C-G	474 (56.1)	551 (59.5)	1.000 (reference)	
G-T-C-C-A	29 (3.5)	54 (5.8)	1.602 (1.003–2.557)	0.047
G-C-C-C-A	7 (0.8)	0 (0.0)	0.057 (0.003–1.008)	0.005
A-T-C-C-G	24 (2.9)	1 (0.1)	0.036 (0.005–0.266)	<0.0001
*VEGF* −1154G>A/−1498T>C/+936C>T/+1612G>A/+1725G>A				
G-T-C-G-G	401 (47.6)	475 (51.3)	1.000 (reference)	
G-C-C-A-A	8 (0.9)	0 (0.0)	0.050 (0.003–0.864)	0.002
A-T-C-G-G	23 (2.7)	1 (0.1)	0.037 (0.005–0.273)	<0.0001
A-T-T-G-G	5 (0.7)	0 (0.0)	0.077 (0.004–1.393)	0.021
*VEGF* −1154G>A/−1498T>C/+1451C>T/+1612G>A/+1725G>A				
G-T-C-G-G	402 (47.6)	473 (51.0)	1.000 (reference)	
G-C-C-A-A	7 (0.9)	0 (0.0)	0.057 (0.003–0.996)	0.005
A-T-C-G-G	28 (3.3)	1 (0.1)	0.030 (0.004–0.224)	<0.0001
*VEGF* −1498T>C/+936C>T/+1451C>T/+1612G>A/+1725G>A				
T-C-C-G-G	421 (49.9)	474 (51.2)	1.000 (reference)	
T-C-C-A-A	32 (3.7)	57 (6.1)	1.582 (1.006–2.487)	0.045
T-T-C-G-G	11 (1.3)	0 (0.0)	0.039 (0.002–0.658)	0.001

Note: CAD, coronary artery disease; OR, odd ratio; CI, confidence interval.

**Table 5 jpm-12-00761-t005:** Haplotype analysis of *VEGF* polymorphisms in CAD patients and controls subjects.

Haplotype	Control Subjects (2*n* = 844)	CAD Patients (2*n* = 926)	OR (95% CI)	*p*
*VEGF* −1154G>A/−1498T>C/+936C>T/+1451C>T				
G-T-C-C	503 (59.6)	605 (65.4)	1.000 (reference)	
G-T-T-C	10 (1.2)	3 (0.3)	0.249 (0.068–0.912)	0.023
G-C-C-C	69 (8.2)	52 (5.6)	0.627 (0.429–0.915)	0.015
A-T-C-C	24 (2.8)	2 (0.2)	0.069 (0.016–0.295)	<0.0001
*VEGF* −1154G>A/−1498T>C/+936C>T/+1612G>A				
G-T-C-G	398 (47.2)	474 (51.2)	1.000 (reference)	
G-C-C-A	13 (1.6)	6 (0.6)	0.388 (0.146–1.029)	0.049
A-T-C-G	23 (2.8)	1 (0.2)	0.037 (0.005–0.272)	<0.0001
A-T-T-G	5 (0.6)	0 (0.0)	0.076 (0.004–1.386)	0.020
*VEGF* −1154G>A/−1498T>C/+936C>T/+1725G>A				
G-T-C-G	480 (56.8)	552 (59.6)	1.000 (reference)	
G-T-C-A	29 (3.5)	55 (5.9)	1.649 (1.035–2.629)	0.034
G-T-T-A	4 (0.4)	0 (0.0)	0.097 (0.005–1.801)	0.047
A-T-C-G	24 (2.8)	1 (0.1)	0.036 (0.005–0.269)	<0.0001
A-T-T-G	6 (0.7)	0 (0.0)	0.067 (0.004–1.191)	0.010
*VEGF* −1154G>A/−1498T>C/+1451C>T/+1612G>A				
G-T-C-G	401 (47.5)	472 (51.0)	1.000 (reference)	
A-T-C-G	28 (3.3)	2 (0.2)	0.061 (0.014–0.256)	<0.0001
*VEGF* −1154G>A/−1498T>C/+1451C>T/+1725G>A				
G-T-C-G	479 (56.8)	552 (59.6)	1.000 (reference)	
G-C-C-A	7 (0.9)	0 (0.0)	0.058 (0.003–1.016)	0.005
A-T-C-G	28 (3.3)	1 (0.1)	0.031 (0.004–0.229)	<0.0001
*VEGF* −1154G>A/−1498T>C/+1612G>A/+1725G>A				
G-T-G-G	501 (59.4)	573 (61.9)	1.000 (reference)	
G-C-A-A	7 (0.9)	0 (0.0)	0.058 (0.003–1.024)	0.005
A-T-G-G	28 (3.4)	1 (0.1)	0.031 (0.004–0.230)	<0.0001
*VEGF* −1154G>A/+936C>T/+1451C>T/+1612G>A				
G-C-C-G	450 (53.3)	520 (56.2)	1.000 (reference)	
G-C-T-G	7 (0.8)	1 (0.2)	0.124 (0.015–1.009)	0.029
*VEGF* −1154G>A/+936C>T/+1451C>T/+1725G>A				
G-C-C-G	534 (63.3)	604 (65.2)	1.000 (reference)	
A-T-C-G	6 (0.7)	0 (0.0)	0.068 (0.004–1.211)	0.011
*VEGF* −1154G>A/+936C>T/+1612G>A/+1725G>A				
G-C-G-G	453 (53.7)	522 (56.3)	1.000 (reference)	
A-C-A-A	1 (0.2)	9 (1.0)	7.810 (0.985–61.920)	0.025
*VEGF* −1154G>A/+1451C>T/+1612G>A/+1725G>A				
G-C-G-G	456 (54.0)	522 (56.4)	1.000 (reference)	
A-C-A-A	1 (0.2)	9 (1.0)	7.862 (0.992–62.320)	0.024
*VEGF* −1498T>C/+936C>T/+1451C>T/+1612G>A				
T-C-C-G	418 (49.6)	474 (51.2)	1.000 (reference)	
T-T-C-G	11 (1.3)	0 (0.0)	0.038 (0.002–0.653)	0.001
*VEGF* −1498T>C/+936C>T/+1451C>T/+1725G>A				
T-C-C-G	498 (59.0)	550 (59.4)	1.000 (reference)	
T-C-C-A	29 (3.4)	57 (6.1)	1.780 (1.120–2.829)	0.014
T-T-C-G	11 (1.3)	3 (0.4)	0.247 (0.068–0.891)	0.021
*VEGF* −1498T>C/+936C>T/+1612G>A/+1725G>A				
T-C-G-G	425 (50.4)	474 (51.2)	1.000 (reference)	
T-C-A-A	31 (3.7)	58 (6.3)	1.678 (1.064–2.645)	0.025
*VEGF* +936C>T/+1451C>T/+1612G>A/+1725G>A				
C-C-G-G	537 (63.6)	600 (64.8)	1.000 (reference)	
C-T-G-G	9 (1.1)	1 (0.1)	0.099 (0.013–0.788)	0.009
T-C-G-G	15 (1.8)	3 (0.4)	0.179 (0.052–0.622)	0.002

Note: CAD, coronary artery disease; OR, odd ratio; CI, confidence interval.

## Data Availability

Not applicable.

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
