# Peer review of "Study of the Association between VEGF Polymorphisms and the Risk of Coronary Artery Disease in Koreans"

_jpm, 2022, doi:10.3390/jpm12050761_

Round 1

Reviewer 1 Report

Ko et al. studied the correlation between VEGF genotypes and CAD in a Korean population. The study is well performed and the manuscript well written. Although the authors find an association between certain genotypes or haplotypes and CAD, I feel the results are quite preliminary, as the physiological consequences of these VEGF genotypes are not investigated.

The manuscript would benefit from expression analyses (mRNA) of the different VEGF polymorphisms and allele combinations to give a hint of the underlying mechanism. Additionally, it would be interesting to measure actual VEGF levels in these patients.

Minor points:

  • Please explain the abbreviation MetS the first time used in the text. The abbreviation may not be familiar to all readers.
  • I feel the introduction would benefit from a clearer rationale. Why do the authors hypothesize that VEGF polymorphisms may contribute to CAD specifically? The connection between the two is not yet clearly described in the introduction.
  • Discussion line 325: The authors state that “These results demonstrate that VEGF haplotypes contribute to the occurrence of CAD.” I feel this statement is too strong at this stage, as the study did not follow up on any mechanisms or physical consequences of the expression of the different polymorphic VEGFs. I suggest to use the following instead: “These results demonstrate that VEGF haplotypes are associated with the occurrence of CAD.”
  • Minor corrections of English language are needed.
  • Tables 2 and 3: The legend titles (i.e. “controls (n=422)”) insinuates that the no. of study participants are reported in brackets in the table, which is not the case. Please clarify this in the tables.

Author Response

Ko et al. studied the correlation between VEGF genotypes and CAD in a Korean population. The study is well performed and the manuscript well written. Although the authors find an association between certain genotypes or haplotypes and CAD, I feel the results are quite preliminary, as the physiological consequences of these VEGF genotypes are not investigated.

The manuscript would benefit from expression analyses (mRNA) of the different VEGF polymorphisms and allele combinations to give a hint of the underlying mechanism. Additionally, it would be interesting to measure actual VEGF levels in these patients.

=> Thank you for critical comments. We tried to augment some contents as you suggested. We revised our insufficient descriptions as follows:

Minor points:

  • Please explain the abbreviation MetS the first time used in the text. The abbreviation may not be familiar to all readers.

=> OK, We added the full name of MetS the first time used in the text. [Materials and Methods section, page5, line210]

  • I feel the introduction would benefit from a clearer rationale. Why do the authors hypothesize that VEGF polymorphisms may contribute to CAD specifically? The connection between the two is not yet clearly described in the introduction.

=> Thank you for comment. We agree for your comments. So, we added the following paragraph to the introduction section.

“Especially, one study showed that circulating levels of total VEGF-A and VEGF-A165b in CAD patients were associated with syntax score, indicating the severity and complexity of CAD [30], and another study showed that VEGF 165b induced a neovascular response in the adventitia, and enhanced intimal thickening through the peri-adventitial collar placement [31]. Cai et al. [32] demonstrated that endothelial progenitor cell proliferation mediated by VEGF and IL-8 secretion is related to cardiac shock wave therapy, and Song et al. [33] showed that VEGF, derived from transplanted Bone marrow mesenchymal stem cell, regulated the expression of miRNAs such as miRNA-23a and miRNA-92a and performed anti-apoptotic effects in cardiomyocytes after MI.” [Introduction section, page2, line 80-88]

  • Discussion line 325: The authors state that “These results demonstrate that VEGF haplotypes contribute to the occurrence of CAD.” I feel this statement is too strong at this stage, as the study did not follow up on any mechanisms or physical consequences of the expression of the different polymorphic VEGFs. I suggest to use the following instead: “These results demonstrate that VEGF haplotypes are associated with the occurrence of CAD.”

=> Thank you for comment. We agree for your comments. We revised the sentence.

“These results demonstrated for the first time that haplotypes of VEGF six polymorphisms and genotype combinations are associated with the occurrence of CAD.” [Discussion section, page12, line371-373]

  • Minor corrections of English language are needed.

=> Thank you for comment. We corrected the English before submission in BioScience.

  • Tables 2 and 3: The legend titles (i.e. “controls (n=422)”) insinuates that the no. of study participants are reported in brackets in the table, which is not the case. Please clarify this in the tables.

=> Thank you for your comments. Sorry for confusing to you. So, we revised the manuscript and described the control group as control subjects and the case group as CAD patients in Table 2, 3.

Reviewer 2 Report

The manuscript submitted to the Journal of Personalized Medicine by Dr. Eun Ju Ko et al. and entitled «Study of the association between VEGF polymorphisms and the 2 risk of coronary artery disease in Koreans» is aimed to the determination of single nucleotide polymorphisms in the VEGF gene involving to angiogenesis, stimulating vascular permeability, endothelial cell proliferation, promoting cell migration and inhibiting of apoptosis. Authors suggested that the VEGF gene polymorphisms and clinical factors may impact coronary artery disease prevalence. The obtained results are valuable for the modern personalized medicine and vascular biology, the presented manuscript is well-designed and written, a large amount of the material was analyzed (463 CAD patients and 422 healthy donors), but there are some that must be solved by the authors before publication.

1. In my opinion, the sixth paragraph (Lines 77-88) in the Introduction section can be transferred to the Discussion section because it contains very specific information about certain polymorphic variants.

2. What do we know about the results of GWAS in the context of coronary artery disease? Authors must shortly summarize the recent data about genetic basis of coronary artery disease in the Introduction section and after that explain why the VEGF gene were selected from the large number of the other genes involved in the pathogenesis od coronary artery disease.

3. What were the other exclusion criteria (age limits, cancer, other comorbidities)? Please add this information in the Materials and methods section.

4. Volumes and concentrations of all reagents using in SNP genotyping must be presented in 2.3. Genotyping section.

5. Why authors selected the SNPs listed in the 2.3. Genotyping section? There are a lot of other SNPs in these genes. The full criteria of SNP selection must be presented.

6. The results of the Kolmogorov-Smirnov test or analogues (access the compliance of data with a normal distribution) must be presented to support selection of two-side t-test (probably the non-parametric Mann-Whitney U-test should be selected?).

7. Considering that the authors used a lot of compared groups in their work, they must perform FDR or Bonferroni corrections to avoid type I errors in null hypothesis testing through conducting multiple comparisons.

8. Did the distribution of genotypes and alleles correspond to the population level according to 1000 Genomes or Ensemble databases?

9. I can suggest authors to additionally perform the MDR (multifactor dimensionality reduction) analysis allowing to study gene-gene interactions and to determine the protective and risk alleles and genotypes combinations.

10. To increase the practical value of the obtained results authors must perform the ROC analysis to access the level of significance of particular genotypes as a marker of coronary artery disease.

11. Authors must discuss the differences in genotype frequency between females with coronary artery disease patients and females from control group.

12. In the Discussion section, authors must indicate the novelty and practical importance of the obtained results and compare their own results to the literature data on the role of VEGF gene polymorphisms in the risk of coronary artery diseases.

Author Response

The manuscript submitted to the Journal of Personalized Medicine by Dr. Eun Ju Ko et al. and entitled «Study of the association between VEGF polymorphisms and the 2 risk of coronary artery disease in Koreans» is aimed to the determination of single nucleotide polymorphisms in the VEGF gene involving to angiogenesis, stimulating vascular permeability, endothelial cell proliferation, promoting cell migration and inhibiting of apoptosis. Authors suggested that the VEGF gene polymorphisms and clinical factors may impact coronary artery disease prevalence. The obtained results are valuable for the modern personalized medicine and vascular biology, the presented manuscript is well-designed and written, a large amount of the material was analyzed (463 CAD patients and 422 healthy donors), but there are some that must be solved by the authors before publication.

=> Thank you for critical comments. We tried to augment some contents as you suggested. We revised our insufficient descriptions as follows:

  1. In my opinion, the sixth paragraph (Lines 77-88) in the Introduction section can be transferred to the Discussion section because it contains very specific information about certain polymorphic variants.

=>Thank you for your comments. We checked it. We added the sixth paragraph in the introduction section as it was necessary to describe why we chose six SNPs in VEGF gene.

  1. What do we know about the results of GWAS in the context of coronary artery disease? Authors must shortly summarize the recent data about genetic basis of coronary artery disease in the Introduction section and after that explain why the VEGF gene were selected from the large number of the other genes involved in the pathogenesis of coronary artery disease.

=> Thank you for critical comments. We added the following paragraph in the introduction section.

“Genetic risk is estimated to account for 40~60% of susceptibility to CAD [13]. Since 2007, genome-wide association studies (GWAS) have found significant associations between 321 chromosomal loci and CAD. The identified chromosomal loci were related to blood pressure, lipid metabolism, adiposity, insulin resistance, neovascularization & angiogenesis, immune response and inflammation, NO-signaling, thrombosis, vascular remodeling [14]. Neovascularization and vascular remodeling is associated with lesion formation, and VEGF is one of the genes associated with neovascularization and angiogenesis [15]” [Introduction section, page2, line 57-64]

"Especially, one study showed that circulating levels of total VEGF-A and VEGF-A165b in CAD patients were associated with syntax score, indicating the severity and complexity of CAD [30], and another study showed that VEGF 165b induced a neovascular response in the adventitia, and enhanced intimal thickening through the peri-adventitial collar placement [31]. Cai et al. [32] demonstrated that endothelial progenitor cell proliferation mediated by VEGF and IL-8 secretion is related to cardiac shock wave therapy, and Song et al. [33] showed that VEGF, derived from transplanted Bone marrow-mesenchymal stem cell, regulated the expression of miRNAs such as miRNA-23a and miRNA-92a and per-formed anti-apoptotic effects in cardiomyocytes after MI” [Introduction section, page2, line 80-88]

  1. What were the other exclusion criteria (age limits, cancer, other comorbidities)? Please add this information in the Materials and methods section.

=> Thank you for critical comments. We checked it and there were no the other exclusion criteria.

  1. Volumes and concentrations of all reagents using in SNP genotyping must be presented in 2.3. Genotyping section.

=> Thank you for critical comments. We added the following paragraph in the 2.3 genotyping section.

"PCR was performed using the 2x h-Taq PCR PreMix (Solgent Corporation, Daejeon, Korea).” [Materials and Methods section, page4, line 165-166]

"Real-time PCR was performed using the 2x Real-time Smart mix (Solgent Corporation, Daejeon, Korea).” [Materials and Methods section, page4, line 170-172]

“Genotyping was performed according to the manufacturer’s protocols.” [Materials and Methods section, page4, line 173-174]

  1. Why authors selected the SNPs listed in the 2.3. Genotyping section? There are a lot of other SNPs in these genes. The full criteria of SNP selection must be presented.

=> Thank you for critical comments. We added the following paragraph in the introduction section.

“The -1154G>A and -1498T>C polymorphisms in the promoter region have been studied in CAD patients of other populations, but not in CAD patients in Korea. And, among the 3'UTR region polymorphisms, +1451C>T, +1612G>A, and +1725G>A, excluding +936C>T, have not been studied in CAD.” [Introduction section, page3, line 107-110]

  1. The results of the Kolmogorov-Smirnov test or analogues (access the compliance of data with a normal distribution) must be presented to support selection of two-side t-test (probably the non-parametric Mann-Whitney U-test should be selected?).

=> Thank you for your comments. We agreed your suggestion and performed Kolmogorov-Smirnov test. And, we revised the Table 1 according to the statistic results and added the following sentence in the materials and methods section.

“And, we checked the statistical normality of the continuous variables by confirming that they matched the normal distribution through the Kolmogorov–Smirnov test. The continuous variables showed non-normal distributions (p < 0.05 in Kolmogorov-Smirnov test), and the Mann-Whitney test was performed for the group difference analyses. [Materials and Methods section, page4, line187-191]

  1. Considering that the authors used a lot of compared groups in their work, they must perform FDR or Bonferroni corrections to avoid type I errors in null hypothesis testing through conducting multiple comparisons.

=> Thank you for your comments. We agreed your suggestion and have added the FDR-P values in Table 2, 3, Table S3 and the following paragraph in the materials and methods section.

“Calculation of the FDR-P is a way to address the problems associated with multiple comparisons and provides a measure of the expected proportion of false-positives among data.” [Results section, page4, line 196-198]

  1. Did the distribution of genotypes and alleles correspond to the population level according to 1000 Genomes or Ensemble databases?

=> Thank you for your comments. We agreed your suggestion and have added Table S2 and the following paragraph in the Results section.

“The allele frequencies of the associated SNP in other populations, according to 1000 genomes data, was compared with the control and CAD patient groups (Table S2).” [Results section, page5, line 221-223]

  1. I can suggest authors to additionally perform the MDR (multifactor dimensionality reduction) analysis allowing to study gene-gene interactions and to determine the protective and risk alleles and genotypes combinations.

=> Thank you for your comments. We agreed your suggestion and have added Table S5 and the following paragraph in the Results section.

“Genotype combination analysis was performed to confirm combined genotype effect of the six SNPs. Prior to genotype combination analysis, MDR was performed on the six SNPs to identify interactions that influence CAD risk. And the three SNPs (VEGF -1154G>A,-1498T>C, and +936C>T) were selected as the best MDR model. In VEGF -1154G>A/-1498T>C/+936C>T genotype combination analysis, the combined genotype of VEGF -1154GA/-1498TT/+936CC and VEGF-1154GA/-1498TC/+936TT results that indicated an association with CAD risk. (AOR=0.224, P=0.027; AOR=0.230, P=0.048, respectively, Table S5).” [Results section, page9-10, line 272-279]

  1. To increase the practical value of the obtained results authors must perform the ROC analysis to access the level of significance of particular genotypes as a marker of coronary artery disease.

=> Thank you for your comments. We performed the ROC analysis, but we do not found the significance result. This study was not proposed to analyze the value of VEGF polymorphisms as a diagnostic marker, but rather to investigate the association between VEGF polymorphisms and CAD prevalence.

  1. Authors must discuss the differences in genotype frequency between females with coronary artery disease patients and females from control group.

=> Thank you for your comments. We agreed your suggestion and have added the following paragraph in the Discussion section.

“Gender difference in the genetic risk of CAD may be more confounded by environmental/lifestyle risk factors [74], and similarly, another study showed that in women the VEGF +2578C/+634C/+936C haplotype was found to be associated with higher carotid IMT, but in men no similar association was found or it was even in the opposite direction [75]. Also, recently, one study performed sex-stratified GWAS, identifying 10 variations that were associated with CAD, and 9 of the 10 had a stronger effects in male participants. Conversely, MYOZ2 rs7696877 polymorphism showed significant effect in females [14].” [Discussion section, page11, line358-365]

  1. In the Discussion section, authors must indicate the novelty and practical importance of the obtained results and compare their own results to the literature data on the role of VEGF gene polymorphisms in the risk of coronary artery diseases.

=> Thank you for your comments. We agreed your suggestion and have added the following paragraph in the Discussion section.

 “Case–control studies have suggested that the -1154G>A, +936C>T polymorphisms of VEGF are associated with susceptibility to CAD [71]. However, the -1154G>A, +936C>T polymorphisms have not been associated with CAD in other studies [72, 73]. Clearly, the results of previous studies between the -1154G>A, +936C>T polymorphism and CAD is controversial.” [Discussion section, page11, line351-355]

“These results demonstrated for the first time that haplotypes of VEGF six polymorphisms and genotype combinations are associated with the occurrence of CAD, and that interactions between VEGF polymorphisms and other clinical factors are associated with the prevalence of CAD.” [Discussion section, page12, line371-374]

We revised some errors and awkward descriptions through revisions. The revised phrases or sentences are labeled by blue color.

Please find the revised manuscript.

Round 2

Reviewer 1 Report

The authors have addressed the points adequately and the manuscript has improved by the changes made. Only minor English errors need to be corrected.